# Small area estimation and childhood obesity surveillance using electronic health records

**Ying-Qi Zhao**[1]*, **Derek Norton**[2], **Larry Hanrahan**[3]

**1** Fred Hutchinson Cancer Research Center, Seattle, WA, United States of America, **2** Department of Biostatistics and Medical Informatics, University of Wisconsin-Madison, Madison, Wisconsin, United States of America, **3** Department of Family Medicine and Community Health, University of Wisconsin-Madison, Madison, Wisconsin, United States of America

* yqzhao@frechutch.org

## Abstract

There is an urgent need for childhood surveillance systems to design, implement, and evaluate interventions at the local level. We estimated obesity prevalence for individuals aged 5–17 years using a southcentral Wisconsin EHR data repository, Public Health Information Exchange (PHINEX, 2007–2012). The prevalence estimates were calculated by aggregating the estimated probability of each individual being obese, which was obtained via a generalized linear mixed model. We incorporated the random effects at the area level into our model. A weighted procedure was employed to account for missingness in EHR data. A non-parametric kernel smoothing method was used to obtain the prevalence estimates for locations with no or little data (<20 individuals) from the EHR. These estimates were compared to results from newly available obesity atlas (2015–2016) developed from various EHRs with greater statewide representation. The mean of the zip code level obesity prevalence estimates for males and females aged 5–17 years is 16.2% (SD 2.72%); 17.9% (SD 2.14%) for males and 14.4% (SD 2.00%) for females. The results were comparable to the Wisconsin Health Atlas (WHA) estimates, a much larger dataset of local community EHRs in Wisconsin. On average, prevalence estimates were 2.12% lower in this process than the WHA estimates, with lower estimation occurring more frequently for zip codes without data in PHINEX. Using this approach, we can obtain estimates for local areas that lack EHRs data. Generally, lower prevalence estimates were produced for those locations not represented in the PHINEX database when compared to WHA estimates. This underscores the need to ensure that the reference EHRs database can be made sufficiently similar to the geographic areas where synthetic estimates are being created.

## Introduction

Worldwide obesity prevalence has essentially tripled in the last 40 years. Nearly 2 billion adults are now overweight, and 650 million of them are obese [1]. Childhood obesity, a predictor of adult obesity [2,3], has also dramatically increased in the United States [4]. Thus a public health strategy of preventing childhood obesity is needed to improve population health. To

**Data Availability Statement:** Data cannot be shared publicly due to concerns of patient privacy. Data are available from the University of Wisconsin–Madison School of Medicine and Public Health IRB for researchers who meet the criteria

for access to confidential data (irbrelaince@wisc.edu).

**Funding:** This work was supported by funding from the National Institute of Child Health and Human Development (grant no. R21HD086754), received by Yingqi Zhao.

**Competing interests:** NO authors have competing interests.

accomplish this, there is an increasing need for comprehensive childhood obesity surveillance systems that can monitor trends and design, implement, and evaluate interventions.

Many factors influence childhood obesity, including age, gender, race/ethnicity, socioeconomic status, caloric intake (amount and types), and physical activity [5]. These factors are contextualized by community assets and the built environment and may explain local community variation in obesity prevalence [6,7]. Therefore, there is a substantial need for local surveillance systems and public health action on childhood obesity.

National datasets, such as the Behavioral Risk Factor Surveillance System (BRFSS) and the National Health and Examination Survey (NHANES), may have limited utility in evaluating obesity at local levels because of their smaller sample size and they typically are designed to provide statewide or nationwide estimates. However Electronic Health Records (EHRs) may be a viable local obesity surveillance platform because of its big data scale [8–10].

Previous studies have shown that in combination with Census data, EHRs can reliably duplicate US childhood [6] and US adult [11] obesity estimates. However, can EHRs also be used to construct reliable obesity estimates at a local level? Electronic Health Records are ubiquitous in healthcare throughout the US, but still less common is their availability for local population health surveillance.

In 2009, the University of Wisconsin Electronic Health Record Public Health Information Exchange (UW eHealth PHINEX) database was created to improve population health and clinical practice [12]. It contained EHR data from a south-central Wisconsin academic healthcare system and it was used to assess EHRs viability for creating nationwide childhood [6] and adult [11] obesity estimates.

In 2018 the Wisconsin Health Atlas (WHA) program released statewide and zip code obesity prevalence data based on a collection of EHRs [13]. Data were obtained from several health care systems to provide wide coverage of most of the state's communities and zip codes. The purpose of the program was to create a surveillance system that combined EHRs and local policy data to "1) target interventions and resources where they are needed most, 2) monitor trends over time at meaningful geographic levels, and 3) evaluate ongoing interventions, policies, and programs intended to improve health." [13].

The co-existence of these two systems (WHA and PHINEX) creates an opportunity to conduct a natural experiment. Can a fairly localized EHRs dataset (PHINEX, N ~44,000) reliably estimate local (zip code) obesity risk for all communities in Wisconsin? This question can now be assessed by comparing PHINEX estimates to a much larger EHRs dataset of local community EHRs observations (WHA).

## Materials and methods

This study was approved by the University of Wisconsin-Madison Minimal Risk Institutional Review Board (M2009-1273). Two data sources were compared: the Clinical Electronic Health Record-Public Health Information Exchange (PHINEX) and the Wisconsin Health Atlas (WHA).

The PHINEX program has been described in detail elsewhere [12]. Briefly, Electronic Health Records (EHRs) from 2007–2012 were extracted from the University of Wisconsin Health system and geocoded to the county, zip code, and census block group. The majority of these data came from south central Wisconsin (Dane county and counties surrounding it).

With funding from the University of Wisconsin Partnership Program, The Wisconsin Obesity Prevention Initiative (OPI) was created in 2014 to address Wisconsin's obesity epidemic [14]. Its surveillance component created the Wisconsin Health Atlas [13]: EHR data of obesity prevalence in zip codes throughout Wisconsin. These data were submitted through the

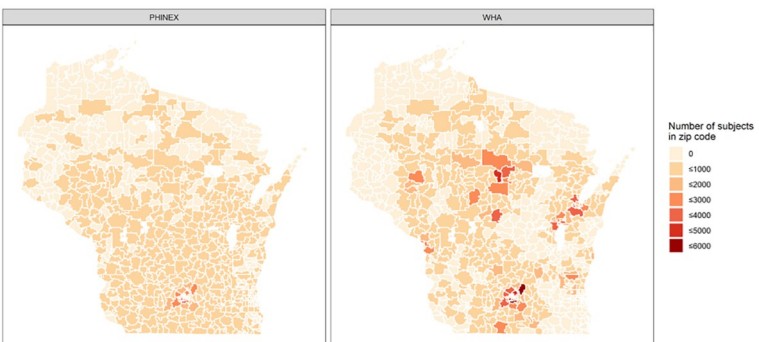

**Fig 1. Data coverages of the State of Wisconsin in UW Public Health Information Exchange (PHINEX) and Wisconsin Health Atlas (WHA).**

Wisconsin Collaborative for Health Care Quality (WCHQ), an organization of 37 healthcare entities that operate within Wisconsin and partners with the program [15]. These local EHRs data cover 2015–2016 and are publicly available on the WHA website [13]. Compared to PHINEX, it has broader geographic coverage throughout the state, and represents information on 226,930 subjects aged 5–17 years. However, there are important geographic areas in WI without data in both data sources, including southeastern WI (i.e. Milwaukee area).

Fig 1 presents data coverages in PHINEX and WHA. It can be seen that the majority of data in PHINEX came from Dane county, but little data in the northern 3$^{rd}$ of the state, and none from the Milwaukee area.

## Subject selection and covariates measures

All subject-level measures from PHINEX were collected from EHRs of subjects with age $\geq 5$ years and $\leq 17$ years as of their most recent BMI data in calendar year 2012. The total number of subjects is 43,752. Individual-level covariates included subjects' sex (male or female), race (white non-Hispanic or non-white / Hispanic), and insurance (none, private, or Medicaid) at the time of the most recent BMI data in 2012. The BMI values were used to assess if a subject was considered obese according to CDC guidelines, with subjects whose BMI values are above the 95$^{th}$ percentile of their BMI-for-age growth charts considered obese [16]. The zip codes of the subjects' addresses were also recorded. There were 43,752 subjects that conformed to the above criteria with complete information. See Table 1 for demographics information.

Three community-level covariates were used in these analyses: urbanicity, economic hardship index (EHI), and similarity index. Urbanicity is based on the 11 Urbanization Summary Groups of Esri's Tapestry which is derived from data on population density, city size, proximity to metropolitan area, and economic/social centrality [17]. For these analyses, urbanicity was categorized as: "urban" for 1–4 (principal urban centers 1&2; metro cities 1&2), "suburban" for 5–8 (urban outskirts 1&2; suburban periphery 1&2), and "rural" for 9–11 (small towns, rural 1 &2). The two designations for principal urban, metro cities, urban outskirts, suburban periphery, and rural areas denote relative affluence within the group [17]. EHI is based on 2007–2011 U.S. Census American Community Survey 5-year estimates and is a measure of community socioeconomic status. EHI scores ranged from 0 to 100, with 100 indicating the highest hardship. It is calculated using the methods of Nathan and Adams [18] normalizing for all Wisconsin census block groups. It incorporates information from six variables: crowded housing (percentage of housing units with more than one person/room), poverty (percentage of households below the federal poverty level), unemployment (percentage of

**Table 1. Demographics in UW Public Health Information Exchange (PHINEX).**

|  | Overall | Male | Female |
|---|---|---|---|
| N | 43752 | 22520 | 21232 |
| Male (N (%)) | 22520 (51.5) |  |  |
| Race/Ethnicity (N (%)) |  |  |  |
| White non-Hispanic | 35738 (81.7) | 18445 (81.9) | 17293 (81.4) |
| Black non-Hispanic | 3111 (7.1) | 1591 (7.1) | 1520 (7.2) |
| Other non-Hispanic | 2174 (5.0) | 1046 (4.6) | 1128 (5.3) |
| Hispanic | 2729 (6.2) | 1438 (6.4) | 1291 (6.1) |
| Age (mean (sd)) | 10.84 (3.78) | 10.76 (3.78) | 10.92 (3.78) |
| Insurance (N (%)) |  |  |  |
| Commercial | 36192 (82.7) | 18630 (82.7) | 17562 (82.7) |
| Medicaid | 7387 (16.9) | 3797 (16.9) | 3590 (16.9) |
| None | 173 (0.4) | 93 (0.4) | 80 (0.4) |
| Obese (N (%)) | 5929 (13.6) | 3359 (14.9) | 2570 (12.1) |

people aged >16 years who are unemployed), education (percentage of people aged >25 years without a high school education), dependency (percentage of the population aged <18 or >64 years), and per capita income. Continuous EHI scores were used in the analysis. The similarity index is derived from nine different community level measures from the US Census Bureaus' 2009–2013 American Community Survey [19]. These nine measures are: percent under 19 years old, percent Hispanic, percent African American, percent of households with >4 occupants, percent in poverty, percent income below $30k, percent with a bachelor's degree, percent employed, and percent employed in management positions.

While urbanicity, EHI, and similarity were all measured at the block group level, use of the similarity index was needed at the zip code level. In order to create a zip code level index, weighted sums of the block group indices were used. For each zip code, the weight assigned to a block group index value was the proportion of that zip code's area that the block group contains (with 0 weight for block groups that do not overlap that zip code). For example, if two block groups overlap a zip code, and the two block groups and comprise 70% and 30% of the zip code's total area, then their respective weights are 0.7 and 0.3, with all other block groups assigned a weight of zero.

## Statistical analysis

We used a two-step procedure to estimate the obesity rate using PHINEX. First, a generalized linear mixed model was utilized to estimate the probability of each individual being obese. Covariates used in our modeling included subjects' age, sex, and race/ethnicity (binary indicator of White non-Hispanic). A random effect of geographic locations (census block group in this instance) was included to account for correlation among the obesity status of different subjects, since individuals within the same geographic location are more likely similar to each other, e.g., due to comparable economic status and sharing the same public facilities. In EHRs, a patient's information is recorded only when he/she visits a clinic. Hence, the collected data are a biased sample of clinic-goers and are subject to missingness of BMI. Thus in the second step, an inverse probability weighting adjustment [20] was adopted, which was easy to implement and naturally accounted for PHINEX patients that were missing when producing the obesity prevalence estimates. Each observed individual with covariates X could represent $1/p(X) - 1$ similar additional subjects who were missing in PHINEX, in addition to himself/herself, where $p(X)$ was the fitted probability that such an individual had BMI not missing using

PHINEX data. We fitted a logistic regression model to the broader PHINEX sample for missing versus present BMI values, using the collected covariates of age, race, insurance, urbanicity, and economic hardship index (as a community-level measure of poverty). Backwards selection was performed on these 5 covariates using a $P<0.05$ retention criteria.

We further employed a kernel smoothing method [21] to obtain prevalence estimates for zip codes without data or with little data (<20 subjects in a zip code). The "distance" between zip codes used in kernel smoothing was measured using the zip code similarity index. Estimates of these little or no data areas are weighted averages of the estimates from more similar areas, where data are observed. The weights are constructed based on the similarity index, where more similar areas have larger weights.

R version 3.5.1 [22] was used for all analyses. Figures are produced using R packages, including *ggplot2* (version 3.2.0) [23], *maps* (version 3.3.0) [24], *maptools* (version 0.9–5) [25], *mapdata* (version 2.3.0) [26]. Regression parameters were estimated via maximum likelihood using *glmer* in the *lme4* (version 1.1–21) package [27]. Kernel bandwidth $h$ was selected using *bw.nrd* on the similarity distances. Variability was estimated through bootstrapping the above process by sampling with replacement at the subject level, and 1000 bootstrap replications were performed. 95% confidence intervals of the estimates were taken as the lower and upper 2.5% percentiles of these bootstraps. Results on obesity prevalence estimates from the above process were compared to the existing results from WHA study using Wisconsin Health Atlas data from 2015–2016, as both results produced obesity prevalence estimates at the zip code level, separately for males and females aged 5–17 years.

## Results

Overall, 47,752 PHINEX patients aged 5 to 17 years (mean 10.8 years; SD 3.8 years) had complete data on covariates during the 2007 through 2012 time period; 13.6% were obese at the most recent BMI reading. The sample was 51.5% male, with race/ethnicity categories of 81.7% White, 7.1% Black, 6.2% Hispanic, and 5.0% other, and insurance categories of 82.7% commercial, 16.9% Medicaid, and 0.4% with no insurance.

The results for the first model fit (generalized linear mixed model) for predicting obesity at the subject level are presented in Table 2. It yielded Odds Ratio (OR) estimates for obesity of 1.04 for a 1-year age increase (95% CI 1.03–1.05), 1.29 for male compared to female (95% CI 1.22–1.37), 1.67 for non-White or Hispanic compared to White (95% CI 1.56–1.80), and 1.72 and 0.843 for Medicaid and No Insurance compared to Commercial Insurance (95% CI's 1.60–1.85 and 0.514–1.31).

In the BMI missingness analysis [6], sex, race/ethnicity, payer, urbanicity, and EHI were all significantly associated (p<0.05) with missing BMI and retained in the model. From this model, patients more likely to have a valid BMI (non-missing BMI) were female compared to male; non-Hispanic Black compared to non-Hispanic White; had insurance (commercial,

**Table 2. Results from generalized linear mixed model for predicting obesity at the subject level.**

|  | Estimate | Std. Error | OR | lower 95% CI | upper 95% CI | p-value |
|---|---|---|---|---|---|---|
| Age | 0.041 | 0.004 | 1.042 | 1.034 | 1.049 | < 0.0001 |
| Gender | 0.257 | 0.029 | 1.293 | 1.223 | 1.368 | < 0.0001 |
| Medicaid insurance | 0.543 | 0.036 | 1.722 | 1.603 | 1.849 | < 0.0001 |
| No insurance | -0.171 | 0.226 | 0.843 | 0.541 | 1.312 | 0.448 |
| White | -0.516 | 0.038 | 0.597 | 0.554 | 0.643 | < 0.0001 |

Note: Reference categories are female, commercial insurance, non-white.

**Table 3. Prevalence estimates in PHINEX and WHA.**

| | PHINEX | | WHA | | Differences in prevalence | |
|---|---|---|---|---|---|---|
| | Mean | SD | Mean | SD | Mean | SD |
| All | 16.2% | 2.7% | 18.3% | 5.9% | -2.1% | 5.4% |
| Male | 17.9% | 2.1% | 19.5% | 5.9% | -1.6% | 5.5% |
| Female | 14.4% | 2.0% | 17.0% | 5.6% | -2.7% | 5.2% |

Medicaid) compared to not having insurance; lived in a suburban versus urban census block group; and lived in an urban versus rural census block group (all $P<0.001$). Children living in areas with higher economic hardship were less likely to have valid BMI recorded ($p<0.001$).

The proposed process yielded mean obesity prevalence estimates at the zip code level for ages 5–17 of 17.9% for males and 14.4% for females, with respective SDs of 2.14% and 2.00%, and respective ranges of 8.5–29.2% and 6.8–25.4%. We compared our results to existing results from WHA study [13]. For the same subject groupings, WHA had mean obesity prevalence estimates of 19.5% for males and 16.9% for females, with respective SDs of 5.86% and 5.59%, and respective ranges of 7.8–45.5% and 6.1–38.2%. Results are presented in Table 3.

The mean widths of the bootstrapped 95% confidence intervals in the proposed process are 4.81% (SD 3.65%) for males and 3.99% (SD 2.87%) for females. For comparison, the mean WHA 95% confidence interval widths are 12.5% (SD 6.40%) for males and 11.7% (SD 6.25%) for females.

When comparing zip-code-by-sex estimates of obesity prevalence from the process in this paper to that of WHA estimates (this paper's estimate minus the WHA estimate), on average, this paper's process produces lower prevalence estimates (2.12% lower on average). Additionally, the range of disagreement between the estimates is quite large, with a difference range of -25.8% to +10.5%, and a difference SD of 5.40%. Fig 2 shows the magnitude of prevalence difference estimates (between the process here and WHA) versus the zip code similarity indices. While there is a positive correlation between the estimates in the two processes, Fig 2 demonstrates that the trend between the two estimates doesn't follow the identity line well. These

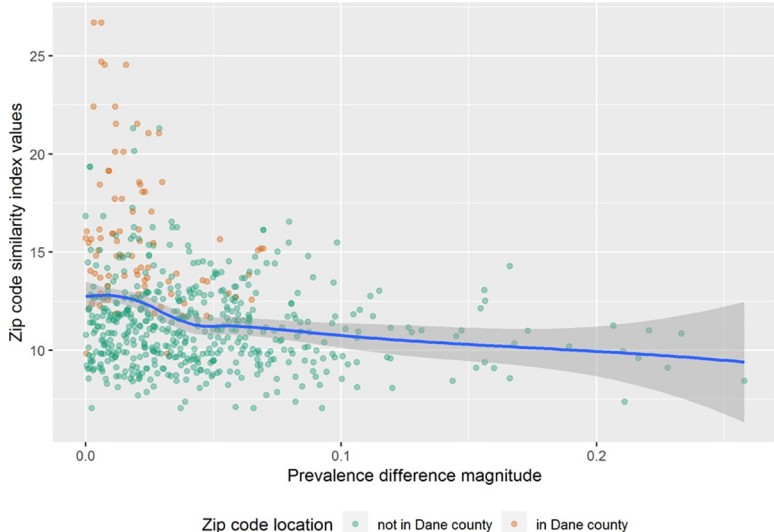

**Fig 2. Scatter plot showing prevalence difference estimates versus zip code similarity indices between Wisconsin Health Atlas and University of Wisconsin Public Health Information Exchange.**

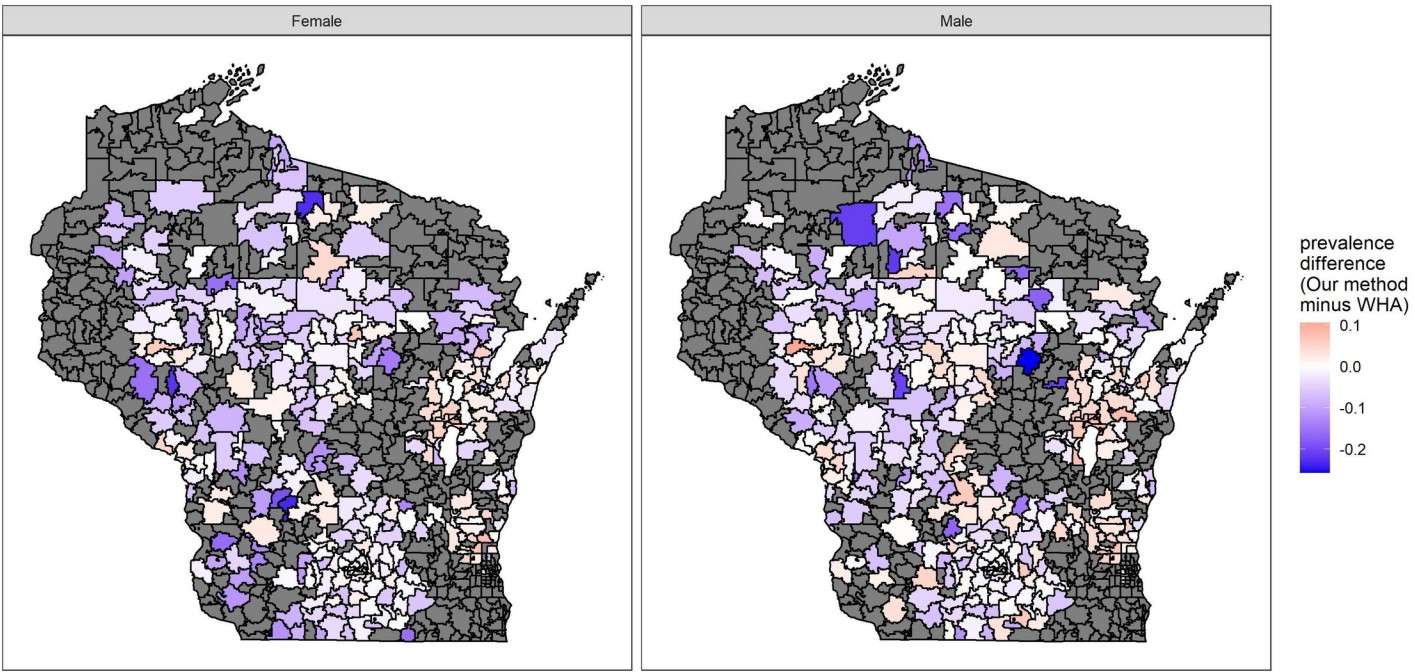

**Fig 3. Differences in prevalence estimates across the State of Wisconsin from PHINEX and WHA (left panel: Male, right panel: Female).**

differences and their variability are much reduced when restricting to the zip codes in Dane county only, with a mean difference of -0.710%, range of -6.96% to +6.38%, and SD of 2.66%. Zip code similarity values from Dane county tend to be larger than non-Dane zip codes (mean similarities of 15.9 vs 11.2). Dane county contains 36 Zip codes. It is the second-most populous county in Wisconsin, where the state capital, Madison, is located. Since PHINEX does not cover data from Milwaukee area (Milwaukee county), Dane county is the largest county with collected data in PHINEX. Hence, it indeed distinguishes itself from the remaining area. Additionally, the estimates between the two processes more closely scatter around the identity line for Dane county zip codes compare to the state as a whole.

In Fig 3, we present the differences in prevalence estimates from PHINEX and WHA. For zip codes where the process in this paper had insufficient raw data (<20 subjects), these estimates were similar in central tendency and distributional shape as the final estimates from zip codes with 20 or more subjects (obesity prevalence estimates' means of 16.2% and 16.1% respectively). Though the variability of the insufficient data areas' estimates is lower than areas with sufficient data: respective SDs of 2.36% and 3.02% and ranges of 9.68–25.0% and 6.79–29.2%.

## Discussion

Despite the widespread adoption of EHRs, resulting in large amounts of health data collected during regular clinical visits, the utilization of EHR data for population health surveillance is still underdeveloped. We propose innovative statistical methods that use EHR data for childhood obesity estimation purposes, in hopes of increasing the use of EHR data and extending the scope of population health surveillance. In particular, to obtain prevalence estimates at a granular level, we used small area estimation techniques, which flexibly incorporate the random effects at the area level. This has been widely employed to generate estimates of health-

related outcomes [28–30]. Moreover, since the EHR samples are collected to document patient-provider interactions, and can be subject to large amounts of missingness, we employed a weighted procedure to handle areas with little or no data.

Compared to existing literature, we further employed a nonparametric kernel smoothing method to obtain the corresponding prevalence estimates for areas with little or no data, based on the estimates from those locations where EHR data is available. This additional step enabled us to utilize local EHR data to estimate childhood obesity prevalence in neighborhoods with little to no data in the EHR. The kernel smoothing step performs better with more data being available and representative of target populations. Otherwise, estimations may not work well, as we see in the discrepancy between PHINEX and WHA prevalence estimates for regions with little or no data. Since we cannot obtain access to the individual level data from WHA, we are limited to compare our results to the existing results in WHA studies, which is a limitation here.

The lower average prevalence estimates in zip codes without data in PHINEX, compared to WHA estimates, could be due to 1) temporal effects as PHINEX data were collected from 2007–2012 and WHA data were collected from 2015–2016. Indeed, national indicators (NHANES) have shown some increases in prevalence rates, with some plateauing. And 2) the majority of data in PHINEX came from Dane county, which has a health ranking of 12[th] out of 72 Wisconsin counties [31], with decreasing amounts of data from zip codes farther from the county. As noted earlier, there is very little data in the northern 3[rd] of the state, and none from the Milwaukee area. Thus, PHINEX data has higher socioeconomic status (SES), better employment, health behaviors, and health outcomes compared to the majority of other Wisconsin counties [31]. On the other hand, WHA has broader data representation (Fig 1), with high data densities in the northern third of the state compared to PHINEX, although WHA also lacks data from the Milwaukee area. Since we have little representation from geographic areas further away from Dane county, our kernel weighted prevalence estimates for those regions with little or no data are generally lower when compared to WHA estimates. We would expect a better agreement between the two systems if more data from regions other than Dane county becomes available, especially if we can integrate data from other systems which cover the areas with lower SES. In general, we suggest that results from multiple methods should be carefully considered. If there are discrepancies between these methods, we should refer to other resources and expert opinions in making decisions.

The information across different regions were represented using the similarity index in our estimation procedure. Though the similarity index "makes no judgement" about a community, and is meant to be used relative to other communities used to construct the measure, similarity values are positively correlated with community features one would typically associate with lower SES, like percentages of households in poverty, percentages of the unemployed, etc. As such, we both expect and observe that Dane county zip codes have lower similarity values due to generally being higher SES that other WI zip codes, and that as similarity increase, we are in zip codes that are less like the PHINEX data areas, and thus we both expect and observe that the agreement between WHA and our process decrease. While we chose to use similarity index, it is straightforward to fit other types of distance metrics into the proposed framework. Hence, if there are other metrics that could reflect the similarity between different locations in terms of obesity prevalence, one can easily tailor the procedure to incorporate other metrics, which could lead to an improved result.

Using EHR data, the methods presented in this paper were used to obtain childhood obesity prevalence estimates at the zip code level, including estimates for areas of little or no data. The methods flexibly allow the user to choose the granularity of the geographic level, and to determine the "distance" measure they feel is best suited for estimating into areas with insufficient

data, whether that be geographic distance, socioeconomic measures, etc. For obesity prevalence estimates, at the zip code level, for males and females aged 5–17 years in WI, comparing these methods to non-model based methods obtained from a larger collection of EHR data, several important observations were noted. The methods in this paper had good agreement with the larger database estimates for areas similar to those represented in the data that our methods used. For areas dissimilar to those in our method's data, estimates appear to be both biased towards lower prevalence estimates compared to the larger database, with more variability in disagreement, especially for zip codes that had insufficient data. The bias direction appears to be a function of the difference between the zip codes with and without sufficient data than a feature of the methods themselves. This underscores the need to ensure that the reference data used for these methods is sufficiently similar to the geographic areas where synthetic estimates are being created.

EHR-based surveillance provides new opportunities and challenges. It could promote collaborations between clinical services, public health officials and other areas, and facilitate timely decision support tools through the EHRs. The sufficiency regarding the population coverage and risk factor information with EHRs remain an issue to ensure representative, population-wide data. These limitations should be investigated and addressed in the future.

## Author Contributions

**Conceptualization:** Ying-Qi Zhao, Larry Hanrahan.

**Formal analysis:** Derek Norton.

**Funding acquisition:** Ying-Qi Zhao.

**Methodology:** Ying-Qi Zhao, Derek Norton.

**Supervision:** Ying-Qi Zhao.

**Writing – original draft:** Ying-Qi Zhao, Larry Hanrahan.

**Writing – review & editing:** Ying-Qi Zhao, Derek Norton, Larry Hanrahan.

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
