## [Decision Letter · Decision Letter 0]

30 Jun 2020

PONE-D-20-13740

Small area estimation and childhood obesity surveillance using electronic health records

PLOS ONE

Dear Dr. Zhao,

Thank you for submitting your manuscript to PLOS ONE. After careful consideration, we feel that it has merit but does not fully meet PLOS ONE’s publication criteria as it currently stands. Therefore, we invite you to submit a revised version of the manuscript that addresses the points raised during the review process.

With respect to the scientific methods and presentation of results, there is clearly a need for more transparent explanation of dealing with missing data, as well as an improved approach to data presentation, namely providing tables with core results and indicated by the reviewers. Overall, limitations must be adressed in more depth.

We look forward to receiving your revised manuscript.

Kind regards,

Hajo Zeeb

Academic Editor

PLOS ONE

Journal Requirements:

'This work was supported by funding from the National Institutes of Health (NIH) (grant nos.

R21 HD086754 and P30 CA01570 ).

'The work is supported by R21HD086754.'

Additional Editor Comments (if provided):

Reviewers' comments:

Reviewer's Responses to Questions

**Comments to the Author**

1. Is the manuscript technically sound, and do the data support the conclusions?

Reviewer #1: Yes

Reviewer #2: Yes

2. Has the statistical analysis been performed appropriately and rigorously? 

Reviewer #1: Yes

Reviewer #2: I Don't Know

3. Have the authors made all data underlying the findings in their manuscript fully available?

Reviewer #1: No

Reviewer #2: No

4. Is the manuscript presented in an intelligible fashion and written in standard English?

Reviewer #1: Yes

Reviewer #2: Yes

5. Review Comments to the Author

Reviewer #1: The present manuscript investigates the use of electronic health records for obesity surveillance, particularly at a regionally small-scale level.

The introduction presents sufficient information to understand how the authors planned to overcome the lack of regional information on obesity prevalence using electronic health records and new datasets to validate HER for surveillance purposes.

Methods are thoroughly described and provide transparent information on scores and necessary steps in data management and statistical analysis. In particular, the two step procedure is an interesting approach to deal with missing data from health records and to estimate the obesity prevalence.

My major concern is the presentation of the results without tables of values or descriptive statistics about Prevalences, differences in prevalence, or Results from the logistics regressions to missingness in BMI. The authors chose to present crude results in Fig2 and Fig1 should be placed in the method section.

First, Fig2 needs slightly more explanation to the legend itself. Second. Fig1 should include some orientation (State within the US, Dane county, Milwauke area etc.) and better highlights the differences in prevalence. This should not be explained in the Discussion, but in the method section. If needed, other statistics of the results might be presented also as a map, but due to the large number of counties (zip codes?) it is hard to verify the picture.

As a rule of thumb, most of the numbers/results presented in the results section should be presented in a table that comprises these interesting results.

The discussion is balanced and highlights benefits and drawbacks of this approach or explains them due to differences in data sources. The authors should close the discussion with one or two general statements on the use of EHR for surveillance that will be kept in mind after reading.

Some minor comments apply.

Line 70 – 73: Please state the number subjects that is mentioned in table 1 also in the first paragraph about PHINEX

Table 1: The table might provide further insights, if descriptive statistics are also presented for age or sex strata.

Line 98 – 101: The categorization of Urbanicity is not clear. Please further explain the 11 Groups which are somehow a score that can be categorized. How is this score roughly derived?

Line 151f.: Please add also the version of package or function that you used in R.

Line 157f.: I might have missed it, but please specify, if you used the similar approach for the data from the Wisconsin Health Atlas for the reader. Or was data sufficient to calculate actual prevalence? I guess not, since these also based on EHR.

Reviewer #2: The authors present a statistical method to estimate obesity prevalence based on EHR data from Wisconsin. The underlying aim is to make more use of existing and collected EHR data. Thus, authors compare their own results with the published Wisconsin Health Atlas estimates. Not a statistician myself, I prefer not to evaluate the methods applied. However, from a non-statistician point of view, the methods seem sound and aspects such as missingness, non-randomness have been addressed, as well as Bootstrapping to account for variability in obesity and non-random data availability. Overall, the paper reads well and is of public health interest questions and has some limitations to be addressed or better discussed.

Major points:

Authors do not discuss the limitations of their study sufficiently, such as the fact that the data they compare do not stem from the same period, nor the origin of the data, essentially EHR in both, but PHINEX from academic centers. The reviewer suggests the authors explain the choice of time periods investigated. The comparability is limited.

Differences in populations in PHINEX and WHA are assumed to be different with respect to SES and health behavior by the authors based on the geographical distribution of respective data. What is the basis of this assumption? Are these differences sufficiently corrected for with the methodology applied?

If I understand correctly, authors did not use WHA data themselves but results of the atlas. Differences in estimates based on time period, geographical distribution and data missingness could be investigated by performing the same methods using WHA data. Please comment.

How was the fitted prevalence of non-missingness estimated, did authors refer to other data on children in Wisconsin to inform the analyses on prevalence of the characteristics?

The method section should include the restricted analyses in Dane county. It is repeatedly singled out in the results and discussion section. Does the Dane county correspond to one zip code? Why is Dane county so relevant, authors should explain in more detail?

Information on zip codes and differences in obesity prevalence could be provided in a table or groups of zip-codes with similar characteristics could be grouped together. Range of differences across ZIP codes is quite large, the figure 1 implies less variability in the PHINEX data estimates. Any methodological explanation? Missingness and little data seems to be higher in areas of high risk populations, please expand on this issue.

The authors conclude, that for similar data the two methods are quite comparable in their estimates, but not where they differ. What is the expected public health impact of one or the other method, considering the bias, provided it were to be used for surveillance?

Figures:

The figure 2 in the paper on page 24 (similarities index and difference magnitude magnitude) does not correspond to the description of figure 2 (differences in estimates of obesity using PHINEX and WHA data) on page 11; on page 12 another figure 2 is introduced, corresponding to page 24.

Minor

Discussion

Pg 13, line 246 add references for health ranking

Pg 13, line 249 “thus it has higher SES…” confusing for reader, replace it with PHINEX

Figure 1 should not follow figure 2 in the order of figures in the manuscript. Also, it is placed in the discussion which the reviewer finds odd.

6. PLOS authors have the option to publish the peer review history of their article (what does this mean?). If published, this will include your full peer review and any attached files.

Reviewer #1: No

Reviewer #2: No

---

## [Editor Report · Decision Letter 1]

9 Feb 2021

Small area estimation and childhood obesity surveillance using electronic health records

PONE-D-20-13740R1

Dear Dr. Zhao,

We’re pleased to inform you that your manuscript has been judged scientifically suitable for publication and will be formally accepted for publication once it meets all outstanding technical requirements.

Kind regards,

Hajo Zeeb

Academic Editor

PLOS ONE
---

## [Editor Report · Acceptance letter]

11 Feb 2021

PONE-D-20-13740R1 

Small area estimation and childhood obesity surveillance using electronic health records 

Dear Dr. Zhao:

I'm pleased to inform you that your manuscript has been deemed suitable for publication in PLOS ONE. Congratulations! Your manuscript is now with our production department. 

Kind regards, 

on behalf of

Prof. Hajo Zeeb 

Academic Editor

PLOS ONE